# Mediterranean Diet Interventions for Depressive Symptoms in Adults with Depressive Disorders: A Protocol for a Systematic Review and Meta-Analysis

**DOI:** 10.3390/ijerph192114437

**Published:** 2022-11-04

**Authors:** Bruno Bizzozero-Peroni, Andrés Godoy-Cumillaf, Rubén Fernández-Rodríguez, Eva Rodríguez-Gutiérrez, Estela Jiménez-López, Frano Giakoni-Ramírez, Daniel Duclos-Bastías, Arthur Eumann Mesas

**Affiliations:** 1Health and Social Research Center, Universidad de Castilla-La Mancha, 16071 Cuenca, Spain; 2Instituto Superior de Educación Física, Universidad de la República, Rivera 40000, Uruguay; 3Grupo de Investigación en Educación Física, Salud y Calidad de Vida, Facultad de Educación, Universidad Autónoma de Chile, Temuco 4780000, Chile; 4Department of Psychiatry, Hospital Virgen de La Luz, 16002 Cuenca, Spain; 5Centro de Investigación Biomédica en Red de Salud Mental (CIBERSAM), Instituto de Salud Carlos III (ISCIII), 28003 Madrid, Spain; 6Faculty of Education and Social Sciences, Universidad Andres Bello, Las Condes, Santiago 7550000, Chile; 7Escuela de Educación Física, Pontificia Universidad Católica de Valparaíso, Valparaíso 2340000, Chile; 8Postgraduate Program in Public Health, Universidade Estadual de Londrina, Londrina 86057-970, Brazil

**Keywords:** adulthood, healthy diet, Mediterranean foods, depression, study protocol, systematic review

## Abstract

The associations between Mediterranean diet (MD) adherence and depression levels have been synthesized from observational studies. However, a systematic review with meta-analysis including randomized controlled trials (RCTs) on this relationship in adults with depressive disorders remains lacking. This protocol was conducted according to the Preferred Reporting Items for Systematic Review and Meta-Analysis for Protocols statement. MEDLINE (PubMed), Cochrane CENTRAL, PsycINFO, Scopus, and Web of Science databases will be systematically searched to identify studies published from database inception up to 30 September 2022. The inclusion criteria will comprise RCTs reporting pre-post changes in depression status (symptoms or remission) after a MD intervention compared to a control condition in adults over 18 years with depressive disorders. Pooled effect sizes and 95% confidence intervals will be calculated using the DerSimonian random-effects model. This study protocol determines the methodological approach for the systematic review and meta-analysis that will summarize the available evidence on the efficacy of MD interventions on depressive symptoms in adults with depressive disorders. The findings from this review may have implications for public mental health programs. The results will be disseminated through peer-reviewed publication, conference presentation, and infographics. No ethical approval will be required since only published data will be used. PROSPERO registration number: CRD42022341895.

## 1. Introduction

Depressive disorders ranked among the top 15 leading causes of burden worldwide in 2019 [1], affecting more than 270 million people [2]. The COVID-19 pandemic has created a high-risk scenario wherein some determinants (e.g., mobility or social interactions) of the cardinal features of all depression phenotypes (i.e., the presence of a sad, empty, or irritable mood and loss of interest or pleasure in life) have been affected [3]. In fact, the prevalence of major depressive disorder increased by 53.2 million additional cases in 2020 [4]. This calls for urgent mitigation strategies to promote mental wellbeing and reduce both the prevalence and burden of depressive disorders as well as their health-related consequences [5].

A complex interplay of genetic, biological, psychological, behavioral, and environmental determinants for depression has been proposed [6,7,8]. In particular, the role of diet in the etiology and course of depression has received more attention in recent years as a modifiable lifestyle factor that could contribute to the treatment of this mental disorder [9,10]. The pathways through which diet influences depression could be related to inflammation, oxidative stress, hypothalamic–pituitary–adrenal axis function, tryptophan–kynurenine metabolism, neurogenesis and brain-derived neurotrophic factor, epigenetics, mitochondrial function, and the gut microbiota [11]. Some evidence suggests that a healthy food matrix [11], such as that of the Mediterranean diet (MD), could influence some of these physiological pathways and, therefore, might potentially prevent the onset and relieve the symptomatology of depression [12,13,14,15]. The MD prototype (i.e., olive oil as the principal source of fat, high consumption of fruits, legumes, nuts, olives, seeds, spices, vegetables and whole-grain cereals, moderate consumption of eggs, fish or seafood, dairy products, red wine and white meat, and low consumption of red or processed meat and sweets) leads to a favorable nutrient intake (i.e., rich in dietary fiber, unsaturated fatty acids, prebiotics, vitamins, polyphenol compounds and carotenoids) associated with reduced oxidative stress [16] and proinflammatory cytokine expression [17] that drive important signaling events in the depression status [13].

The prospective cohort associations between MD adherence and depression in adults have been synthesized in several systematic reviews and meta-analyses of prospective observational studies, which showed mixed results [10,18,19,20,21]. While high MD adherence reduced the incidence of depression over time in some of these evidence syntheses [10,18,19], no significant association was observed in two other reviews [20,21]. Although observational studies provide relevant evidence, randomized controlled trials (RTCs) are known to provide a higher level of evidence for validating and judging the effects of an intervention (such as MD) on a health outcome (such as depressive symptomatology) [22]. Particularly, only RCTs can overcome the epistemological problems of dietary measurement error and form a sounder basis for informing dietary recommendations in human nutrition [23]. Previous systematic reviews of RCTs have analyzed the diet–depression association [24,25,26] but did not specifically examine MD as the exposure [25,26], did not consider adults with a diagnosis of depression [24,25], and did not perform a meta-analysis [24,25,26]. Moreover, in a preliminary search, new evidence from RCTs [27,28,29] was found, indicating that an update is required to help clinicians determine the extent to which MD should be recommended in the treatment of patients with depressive disorders. Therefore, the aim of this protocol is to provide a detailed plan to conduct a systematic review and meta-analysis synthesizing the available evidence from RCTs regarding the effectiveness of MD interventions on depression symptoms in adults with depressive disorders.

## 2. Materials and Methods

This protocol was reported in accordance with the Preferred Reporting Items for Systematic reviews and Meta-Analyses for Protocols (PRISMA-P) statement [30] (Appendix A). The systematic review and meta-analysis will be conducted according to the PRISMA 2020 guidelines [31] and following the Cochrane Collaboration Handbook for Systematic Reviews of Interventions [32]. The overall content of the present protocol is registered in PROSPERO (CRD42022341895).

### 2.1. Eligibility Criteria

The rationale for the eligibility criteria was performed using the patient, intervention, comparison, outcome, and study design (PICOs) framework [32]. To be included, studies retrieved from the peer-reviewed literature must report the following: (i) population: adults over 18 years of age with a diagnosis of depressive disorder; (ii) intervention: experimental strategy that follows a Mediterranean dietary pattern, such as MD-related dietary advice or cooking workshops; (iii) comparator: control condition such as habitual diet, usual care, or ‘social’ groups; (iv) outcome: changes in depression status (i.e., number or severity of symptoms, or remission) for intervention and control groups separately; and (iv) study design: peer-reviewed RCTs.

Moreover, studies will be excluded if they report results in (i) population: individuals with different mental health disorders (e.g., depression, anxiety, bipolar disorder) without stratified data for those with only depression and participants under 18 years of age, even though the mean age of the study sample is over 18 years; (ii) intervention: diet in terms of intake of single nutrients, food items, and food groups; (ii) outcome: depressed mood and depression data that cannot be isolated and extracted (e.g., bipolar disorders, overall mood states, psychological stress); and (iii) data published as conference/meeting abstracts.

#### 2.1.1. Population

We will consider the entire adult population (18 years or more) with a diagnosis of depressive disorders. For this study, RCTs that enrolled subjects with depressive disorders (i.e., major depressive disorder including major depressive episode, persistent depressive disorder, premenstrual dysphoric disorder, substance/medication-induced depressive disorder, depressive disorder due to another medical condition, other specified depressive disorder, and unspecified depressive disorder) according to the fifth edition of the Diagnostic and Statistical Manual of Mental Health [3] will be included. Additionally, RCTs that determined the presence of a depressive disorder through trained professionals using a recognized diagnostic schedule, self-report of medical diagnosis or antidepressant treatment, or validated rating scale to specify high levels of depressive symptomology will be considered. No restrictions in terms of health status or sociodemographic characteristics will be imposed.

#### 2.1.2. Intervention

The intervention under investigation will involve any type of face-to-face or virtual experimental treatment (e.g., dietary advice, cooking workshops, individual or group therapy sessions, provision of relevant Mediterranean foods) directly related to the MD; moreover, studies that implemented MD interventions along with a lifestyle or nutrient supplementation program will be included.

An important issue in nutrition research is the lack of detail and the great variability in the definitions of interventions [33]. The label MD is a concept widely used to describe the food prototype of the Mediterranean basin; however, its key determinants differ, even among experts [34]. Therefore, MD interventions will require at least two key items to reach a Mediterranean food matrix [35]. First, high consumption of plant-based foods, particularly, fruits, vegetables, and legumes; second, a high monounsaturated/saturated fat ratio (high consumption of food sources with high amounts of monounsaturated fat such as nuts, olive oil or whole grains and low consumption of food sources with high amounts of saturated fat such as butter, pastries, or processed meat). The rationale for this focus has two explanations. On the one hand, it is based on previous studies [36,37], which highlight the protective effects of MD. These benefits might be most attributable to specific food sources, such as fruits, legumes, nuts, olive oil, vegetables, and whole grains. Alternatively, it will cover the main MD components without neglecting the possible adaptations of this dietary pattern to the geographical location and sociocultural customs. Although the discrepancies in the MD definition could lead to inconsistencies and bias in the intervention-outcome relationship [33], this study will include RCTs from non-Mediterranean regions where this food system model (i.e., maintaining the abovementioned key components) can be adapted to country-specific agricultural resources and cultures and may have health benefits similar to those reported in Mediterranean regions [38].

#### 2.1.3. Comparison

The review will include RCTs that present as comparators, control groups such as usual care, ‘social’ groups (e.g., befriending support sessions), standard diet, dietary advice unrelated to MD, or other diet regimens. All comparators will be recorded when randomization and the completion of outcomes are reported, as required for RCTs.

#### 2.1.4. Outcome

Depression outcomes will be defined as changes in depressive symptoms according to validated [39] observer rating scales and self-rating scales (continuous data collected by screening instruments such as the Beck Depression Inventory, Hamilton Depression Rating Scale, Inventory of Depressive Symptomatology, Montgomery–Asberg Depression Rating Scale or Zung Self-Rating Depression Scale) and depression remission (dichotomous data defined by a prespecified threshold on a depression scale or no longer meeting clinical criteria for depression). Depressed mood and depression data that cannot be isolated and extracted (e.g., bipolar disorders, overall mood states, psychological stress) will not be included as depression outcomes.

#### 2.1.5. Study Design

Only RCTs will be included because they provide (specifically high-quality RCTs) the most reliable evidence for healthcare intervention and clinical decision-making [22,23].

### 2.2. Search Methods for Study Identification

The systematic search will be conducted in MEDLINE (PubMed), Cochrane CENTRAL, PsycINFO, Scopus and Web of Science from database inception until 30 September 2022. Additional searches will be performed in the International Clinical Trials Registry Platform and the ClinicalTrials.gov websites to capture any study not covered by the main search. Further studies will be identified by screening the reference lists of the included studies and relevant reviews for potential relevance. The authors will be contacted in case of a lack of data. The electronic database searches will be limited to keywords, title, and abstract. The search strategy (Table 1) will involve a set of free text terms grouped from the PICO strategy. The search will first be carried out in MEDLINE and will subsequently be adapted to the other databases. No limits will be applied to the study language. In the case of non-English language articles, Google translate will be used to assess the eligibility of article abstracts, and those studies deemed eligible for full-text assessment will be translated by nonprofessional translators (i.e., volunteer researchers), by the University language department, or by professional translators who are native speakers of the language if necessary.

### 2.3. Data Collection and Analysis

#### 2.3.1. Study Selection

All files of references from databases and other repositories will be imported into Mendeley Manager (v1.19.8; Elsevier, London, UK). Duplicate references will be identified using the in-built Mendeley function “check for duplicates”. Following this step and based on inclusion/exclusion criteria, two researchers will first independently examine the titles and abstracts and then screen the full text of the studies identified, with consensus required for final inclusion. Discrepancies will be resolved by consulting a third reviewer. The results of searches in databases and other repositories and the selection process will be displayed using the PRISMA 2020 flow diagram (Figure 1).

#### 2.3.2. Data Collection Process

Two researchers will independently extract data extraction from the included studies on a standardized template, where the accuracy will be checked in case of discrepancies. If necessary, additional data will be requested from the corresponding authors via email.

The following study-specific data will be extracted: (1) name of the first author and year of publication; (2) country; (3) study design; (4) sample size; (5) characteristics of the participants (age, sex, type and duration of depressive disorder, medications); (6) MD intervention characteristics (type, duration, dietary components, adherence to the intervention according to follow-up measures, such as food diaries or monitoring face-to-face sessions, and pre-post adherence to MD according to diet quality measures, such as self-report questionnaires or spectrophotometer); (7) measures of depression; and (8) main findings. The information will be summarized in a “table of characteristics” (Appendix A).

#### 2.3.3. Risk of Bias

The risk of bias will be independently assessed by two researchers using the Cochrane Collaboration tool [40]. In cases of discrepancies, a third reviewer will be consulted.

#### 2.3.4. Certainty of the Evidence

The Grading of Recommendations Assessment, Development and Evaluation (GRADE) approach will be used to assess the evidence quality and to provide recommendations [41]. The GRADE method will be applied following five distinct steps: (1) assign an a priori ranking of ‘high’ to RCTs; (2) ‘downgrade’ or ‘upgrade’ the initial ranting; (3) assign final grade for the quality of evidence as ‘high’, ‘moderate’, ‘low’, or ‘very low’ for all critically important outcomes; (4) address other influencing factors that affect the recommendation strength of a course of action; (5) make a ‘strong’ or ‘weak’ recommendation [42].

### 2.4. Synthesis of Data

Once the primary data of the included studies had been extracted, pre vs. post MD interventions will be compared to control conditions for depression levels (i.e., symptoms or remission), and these data will be synthesized narratively. A meta-analysis will be conducted when we identify a minimum of five studies addressing the same outcome [43]. Continuous data (*n*, means, and standard deviations) will be collected from both the intervention and control groups at baseline and at the end of the follow-up period. When clinical trials report other descriptive (e.g., medians and interquartile ranges) or association (e.g., correlation or regression coefficients), these data will be extracted and subsequently converted into standardized measures (i.e., mean or standardized mean difference) that will allow comparison with the measures provided by the other studies [32]. The effect size (ES) and the 95% confidence interval (95% CI) will be calculated for each included study using Cohen’s d index [44]. The pooled effect size (p-ES) for the effect of MD intervention vs. the control condition will be estimated for the RCTs using the DerSirmonian and Laird random-effects method [45,46]. The heterogeneity between the studies will be evaluated using the I^2^ statistic, categorized as not important (0–30%), moderate (30–60%), substantial (60–75%), or considerable (75–100%) [32]. Additionally, the corresponding *p* values and 95% CIs for I^2^ will be considered [47].

If there is available information, subgroup analyses will be performed based on characteristics of the population (age, sex, and type of depressive disorder), intervention (type, i.e., dietary treatment only vs. lifestyle program, and duration), comparison (type of control condition, i.e., active—such as befriending support sessions—vs. passive—such as usual diet), outcome (measures of depressive symptoms), and studies (methodological quality). Random-effects meta-regression models will be conducted to consider potential main factors of heterogeneity in the MD-depression association (e.g., age, sex, BMI, total energy intake, medications). To evaluate the robustness of the p-ESs and detect whether any specific study represents a large proportion of heterogeneity, sensitivity analyses will be performed by excluding the studies one by one. In addition, where possible, a sensitivity analysis according to the RCT data analysis method (i.e., completer, per-protocol, intent-to-treat, and modified intent-to-treat analyses) will be considered. Finally, publication bias will be evaluated through visual inspection of funnel plots and Egger’s regression asymmetry test for assessing small study effects (*p* values < 0.05 implicates publication bias) [48].

We will conduct all statistical analyses in R software (version 4.2.1; R Foundation for Statistical Computing).

## 3. Ethics and Dissemination

Ethics committee approval and informed consent from patients will not be required. The planned systematic review and meta-analysis will have public health implications by providing updated evidence on the efficacy of MD interventions on depressive symptoms in adults with depressive disorders. The results of the systematic review and meta-analysis will be submitted to a peer-reviewed journal.

## 4. Discussion

This protocol describes the methodology that will be used for a systematic review and meta-analysis that will synthesize the efficacy of Mediterranean dietary interventions in improving levels of depression (i.e., symptoms or remission) in adults with depressive disorders. Moreover, the systematic review will intend to provide evidence of the main factors related to depression, such as age and sex.

Available evidence indicates that MD is one of the healthiest and most environmentally sustainable dietary matrix patterns [49], promoting a natural, simple, and feasible approach [34] with the potential to prevent the onset and reduce the symptomatology of depression [14,15]. Indeed, high MD adherence has been associated with a reduced risk of depression throughout adulthood in some meta-analyses of prospective cohort studies [10,18,19]. A previous systematic review of RCTs that specifically analyzed MD as a treatment for depressive symptoms in adults with depressive disorders showed both that this dietary pattern would provide a potential therapeutic intervention and the urgent need for more RCTs [24]. Since then, other intervention-controlled trials have been published [27,28,29] that may add important evidence to validate MD as an effective strategy to improve the symptoms of depression. To date, a meta-analysis has not been performed, highlighting the demand for MD recommendations based on the highest level of evidence.

A meta-analysis of multiple RCTs (preferably meticulous, large, and long-term) provides the best opportunity for reliable answers in human nutrition science [23,33], particularly to extrapolate the therapeutic potential of the MD on depression symptoms in adults with depressive disorder. To the best of our knowledge, an updated synthesis of RCTs with a meta-analytical approach is needed to answer the following question: does the available evidence support the efficacy of MD interventions in reducing levels of depression in adult patients with depressive disorder? Since both global dietary transitions [50] and the increasing trend of depression [4] have become growing challenges and public health issues, this study may have future implications for public mental health policies due to the urgent need for effective strategies for the treatment of depression.

The limitations of the review may include the usual shortcomings of systematic reviews and meta-analyses, such as publication bias, low methodological quality, and heterogeneity of the included studies. Differences regarding sample characteristics, dietary interventions, depressive symptoms assessment methods and methodological quality may limit the extrapolation of the findings.

## 5. Conclusions

This study facilitates the protocol methodology for a systematic review and the first meta-analysis of RCTs that will synthesize the efficacy of MD interventions on depressive symptoms in adults with depressive disorders. Specifically, this study will provide an update on the essential evidence regarding the role of MD as a potential nonpharmacological approach to improve the depression status of adult patients with depressive disorder. Findings from this systematic review and meta-analysis will be a key tool for evidence-based decision-making and will be useful for researchers and health professionals responsible for promoting healthy lifestyles and mental health care. The results obtained will be disseminated through peer-reviewed publications, national and international conferences, social networks, educational talks, and infographics.

## Figures and Tables

**Figure 1 ijerph-19-14437-f001:**
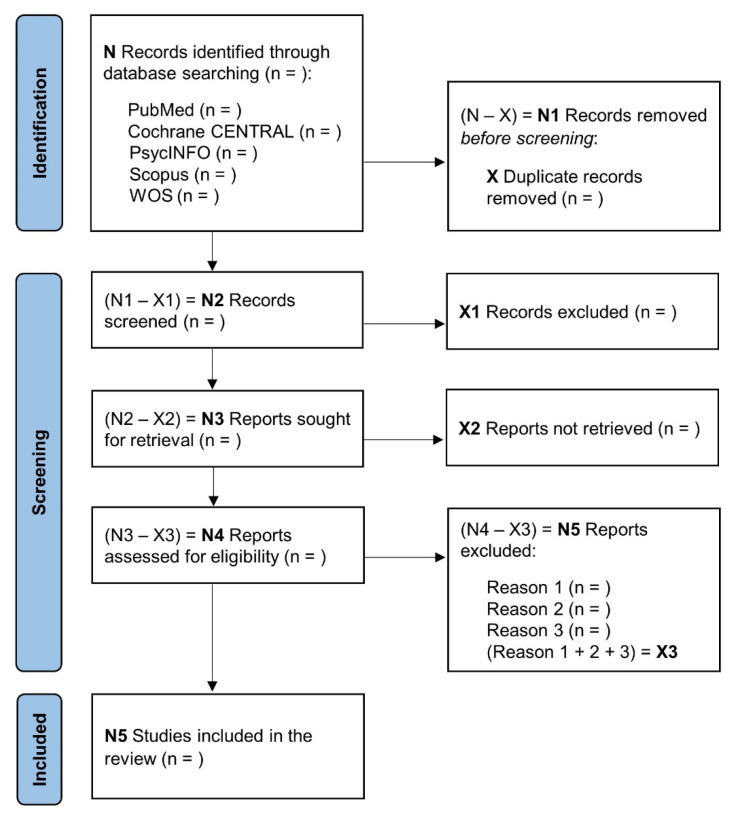
PRISMA flow diagram for identifying, screening, and determining the eligibility and inclusion of studies.

**Table 1 ijerph-19-14437-t001:** Search strategy for the MEDLINE database.

PICO Component	Keywords
**#1 Population**	Adult* OR “young adult” OR “middle aged” OR aged OR elderly OR olde* OR patient
**#2 Intervention**	“Mediterranean diet” OR “Med-Diet” OR “Mediterranean-style diet” OR “Mediterranean food” OR “dietary pattern” OR “diet quality” OR “healthy diet” OR “diet intervention” OR “diet improvement” OR “food therapy”
**#3 Outcome**	Depress* OR dysthymi* OR “dysphoric disorder” OR “mood disorder” OR “affective disorder” OR “affective symptoms”
**#4 Study design**	(random* OR clinical OR controlled OR intervention* OR experimental) AND (trial OR study OR allocation)
**Search Strategy**	[(**#1**) AND (**#2**) AND (**#3**) AND (**#4**)]

Proximity operators (*) will be used to search for root words.

## Data Availability

Not applicable.

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
