# Peer review of "Mediterranean Diet Interventions for Depressive Symptoms in Adults with Depressive Disorders: A Protocol for a Systematic Review and Meta-Analysis"

_ijerph, 2022, doi:10.3390/ijerph192114437_

Round 1
Reviewer 1 Report
The authors have done a great job. It is a protocol for a systematic review and meta-analysis of the relationship between the Mediterranean diet and depressive symptoms in adults who previously had depression. To this end, the authors have made a thorough introduction on the current problem of depression, increased by the pandemic. In addition, they also discussed the importance of the Mediterranean diet and how diet could influence the hypothalamic-pituitary-adrenal axis, influencing neurotrophic factors, among others.On the other hand, in the section on materials and methods they have described, following PRISMA standards, all the sections pertaining to the work they are going to carry out, clearly describing the search strategy used and the extraction and synthesis of data.
It is, therefore, a very well elaborated protocol.
Minor revisions:
- How are they going to measure adherence to the Mediterranean diet? This should be included in section 2.3.2.
- I would advise the authors to include in section 2.4. p value they are going to consider for Egger's test.
Congratulations on the work.
Reviewer 2 Report
This protocol details a systematic review and meta-analysis to be undertaken on Mediterranean diet interventions for depressive symptoms in people with a depressive disorder. The manuscript is largely well written and will be of interest to potential readers. I have detailed some comments below that I believe are critical that have been overlooked in the planned approach.
1. The current PICOS is not quite replicable. Some additional information/clarification could be added. For example, studies that include people under 18 years but the mean age of the study group is 20 years, is this included or not? If a study has predominantly 75% depressive disorder and 25% anxiety disorder, is it included or not? If the intervention includes nutritional supplements in addition to nutrition advice/consultations, is that included or not? Is there any limitation on face-to-face versus tele-health/virtual delivery? I would also note that a control condition that matches the social interaction component of the intervention is highly important to include as a control condition (and can be further explored compared to usual care/habitual diet in the subgroup analysis).
2. The authors state that no restriction will be place on publication language. I would suggest adding how studies in other languages will be included, i.e., consulting someone proficient in that language or use of a language translation program.
3. To be able to answer the question sought for this study, authors should be extracting information related to change in depressive symptoms related to change in dietary intake or other measures of dietary adherence e.g., biochemical measures, e.g., carotenoids. This does not appear to be a consideration at all. It will be easy to argue that other factors are impacting on reduced depressive symptoms without this information, e.g., social interaction of the consultation if the control group design is inappropriate, e.g., simply habitual diet or usual care.
4. The authors present a fairly simple plan of data extraction and analysis. What if the data is not presented as mean and SD? How will the authors adjust for ITT compared to completer analysis approaches?
